# In Vitro-In Silico Tools for Streamlined Development of Acalabrutinib Amorphous Solid Dispersion Tablets

**DOI:** 10.3390/pharmaceutics13081257

**Published:** 2021-08-13

**Authors:** Deanna M. Mudie, Aaron M. Stewart, Jesus A. Rosales, Molly S. Adam, Michael M. Morgen, David T. Vodak

**Affiliations:** Global Research & Development, Lonza, Bend, OR 97703, USA; aaron.stewart@lonza.com (A.M.S.); rosaleja@uw.edu (J.A.R.); molly.adam@lonza.com (M.S.A.); michael.morgen@lonza.com (M.M.M.); david.vodak@lonza.com (D.T.V.)

**Keywords:** acalabrutinib, amorphous solid dispersion, bioavailability enhancement, acid reducing agent, proton pump inhibitor, kinase inhibitor, in silico prediction, absorption modeling, spray drying, GastroPlus

## Abstract

Amorphous solid dispersion (ASD) dosage forms can improve the oral bioavailability of poorly water-soluble drugs, enabling the commercialization of new chemical entities and improving the efficacy and patient compliance of existing drugs. However, the development of robust, high-performing ASD dosage forms can be challenging, often requiring multiple formulation iterations, long timelines, and high cost. In a previous study, acalabrutinib/hydroxypropyl methylcellulose acetate succinate (HPMCAS)-H grade ASD tablets were shown to overcome the pH effect of commercially marketed Calquence in beagle dogs. This study describes the streamlined in vitro and in silico approach used to develop those ASD tablets. HPMCAS-H and -M grade polymers provided the longest acalabrutinib supersaturation sustainment in an initial screening study, and HPMCAS-H grade ASDs provided the highest in vitro area under the curve (AUC) in gastric to intestinal transfer dissolution tests at elevated gastric pH. In silico simulations of the HPMCAS-H ASD tablet and Calquence capsule provided good in vivo study prediction accuracy using a bottom–up approach (absolute average fold error of AUC_0-inf_ < 2). This streamlined approach combined an understanding of key drug, polymer, and gastrointestinal properties with in vitro and in silico tools to overcome the acalabrutinib pH effect without the need for reformulation or multiple studies, showing promise for reducing time and costs to develop ASD drug products.

## 1. Introduction

Oncology is the top therapeutic area in pharmaceutical drug development, accounting for 25% of drugs approved by the FDA over the last decade [1]. However, many oncology drugs are poorly water soluble across at least part of the gastrointestinal (GI) pH range, often manifesting as drug–drug or food–drug interactions that can limit oral bioavailability [2]. Calquence^®^ (crystalline acalabrutinib) is an example of an oral oncology drug that exhibits pH-dependent absorption, whereby the area under the plasma drug concentration–time curve (AUC) is reduced by 43% when taken with a proton pump inhibitor (PPI), which elevates gastric pH [3,4]. This drug–drug interaction (DDI) can result in decreased patient compliance and efficacy, since many cancer patients are prescribed PPIs and other gastric acid reducing agents (ARAs) [5].

The mechanism of reduced absorption of Calquence when taken with ARAs is decreased solubility at elevated pH, which is a common mechanism for Biopharmaceutics Classification System (BCS) 2 weak base drugs such as acalabrutinib [6,7]. A previous publication from our laboratory demonstrated that amorphous solid dispersion (ASD) tablets overcame the ARA effect at the human-prescribed 100 mg dose in a fasted beagle dog model [8]. Results demonstrated that ASD tablets achieved similar AUC values at low and high gastric pH conditions and outperformed Calquence capsules 2.4-fold at high gastric pH.

The improved performance of the ASD tablet compared to Calquence was driven by the increased aqueous solubility of the amorphous compared to the crystalline form, which is enough to provide rapid and high extent of GI dissolution at high and low gastric pH values (i.e., in the presence and absence of ARAs). In addition, ASD tablets were 60% smaller than Calquence capsules and showed good physical stability, chemical stability when stored desiccated or refrigerated, and manufacturability as described by Mudie et al. [8]. This outcome highlighted the utility of ASD dosage forms for improving the performance of the numerous weak base and oral oncology drugs that show decreased absorption when taking ARAs [3,9,10].

While the previous publication focused on the in vivo study outcome, the current publication describes a streamlined approach to develop ASD tablets, including formulation selection, in vitro dissolution testing, and in silico simulations used to build confidence in the ASD tablet, overcoming the ARA effect observed with Calquence. A priori in silico plasma concentration–time profiles are compared with in vivo plasma profiles, and in silico prediction accuracy is calculated. This paper describes:Dispersion polymer screeningSpray drying and characterization of ASD intermediatesIn vitro dissolution testing of ASD intermediates versus crystalline acalabrutinibIn vitro dissolution testing of lead 50/50 acalabrutinib/HPMCAS-H ASD tablet versus commercially available CalquencePhysicochemical property measurements of lead ASD intermediateIn silico predictions of in vivo performance

## 2. Materials and Methods

### 2.1. Material Sourcing

Acalabrutinib (CAS 1420477-60-6, >98% purity) was purchased from LC Laboratories (Woburn, MA, USA). Form I was prepared by recrystallizing the purchased acalabrutinib according to WO 2017/002095 Al, Example 1 [11]. See Section A.1 for Form I verification. Hydroxypropyl methylcellulose acetate succinate (HPMCAS) (Aqoat, HF grade, MF grade and LF grade) was purchased from Shin-Etsu Chemical Co., Ltd. (Tokyo, Japan). Poly(methyl methacrylate-co-methacrylic acid (Eudragit L100^®^) was purchased from Evonik (Evonik Industries AG, Essen, Germany). Vinylpyrrolidone-vinyl acetate copolymer (PVPVA) (Kollidon^®^ VA64) was purchased from BASF (Ludwigshafen, Germany). Polyvinylpyrrolidone (PVP) (Kollidon^®^ 30) was purchased from BASF. Hydroxypropyl methylcellulose (HPMC) (Methocel^TM^ E3 LV) was purchased from DuPont de Nemours, Inc. (Wilmington, DE, USA). Methocel^TM^ A4M was purchased from ThermoFisher Scientific (Waltham, MA, USA). Sodium acetate, sodium phosphate, potassium phosphate, hydrochloric acid (HCl), and sodium chloride (NaCl) were purchased from Sigma Aldrich Chemical Company (St. Louis, MO, USA). Fasted-state simulated intestinal fluid (FaSSIF) powder was purchased from Biorelevant.com Ltd. (London, UK). Methanol (HPLC grade) was purchased from Honeywell (Morris Plains, NJ, USA). Tetrahydrofuran (THF) (Optima grade) was purchased from Thermo Fisher Scientific (Waltham, MA, USA). Calquence capsules were purchased from Drug World (Cold Spring, NY, USA). Avicel PH-101 (microcrystalline cellulose) was purchased from FMC Corporation (Philadelphia, PA, USA). Pearlitol 25 (mannitol) was purchased from Roquette America (Geneva, IL, USA). Ac-Di-Sol (croscarmellose sodium) was purchased from Dupont (Wilmington, DE, USA). Cab-O-Sil M5P (fumed silica) was purchased from Cabot Corporation (Alpharetta, GA, USA). Magnesium stearate was purchased from Macron Fine Chemicals/Avantor (Radnor, PA, USA).

### 2.2. Dispersion Polymer Screening

A polymer screening test of seven different polymers was performed using a solvent shift method to assess the polymers’ abilities to sustain acalabrutinib supersaturated drug concentrations [12,13]. The polymers tested included HPMCAS-H, HPMCAS-M, HPMCAS-L, Eudragit L100, HPMC E3, PVP K30, and PVP-VA64. A 20 mg/mL acalabrutinib stock solution (95/5 (*w/w*) tetrahydrofuran (THF)/water) was delivered into an aqueous buffer containing 200 µg/mL of a dissolved polymer using a calibrated pipette. The aqueous buffer consisted of 67 mM phosphate containing 0.5% (*w/w*) (6.7 mM) FaSSIF powder and 82 mM NaCl at a pH of 6.5. A total of 200 µL stock solution was added to achieve a target final concentration of 400 µg/mL acalabrutinib.

During addition of the stock solution, light scattering was monitored by Pion Rainbow™ (Pion Inc., Billerica, MA, USA) ultraviolet (UV) probes (2 mm path length) at 500 to 600 nm (outside the UV absorbance range of acalabrutinib) to observe light scattering with increasing concentration. The scatter signal was corroborated by monitoring direct UV absorbance from 370 to 374 nm to verify that the light-scattering signal could be attributed to crystallization (loss of UV absorbance).

### 2.3. ASD and ASD Tablet Manufacturing and Characterization

Six different acalabrutinib ASDs were prepared using three different grades of HPMCAS (HPMCAS-H, HPMCAS-M, and HPMCAS-L) and two different drug loadings (25 and 50 wt %). HPMCAS-H and HPMCAS-M were chosen since they performed best in the polymer screening study. HPMCAS-L was also selected to determine the impact of HPMCAS grade on relative dissolution and precipitation rates of acalabrutinib. Specifically, in progressing from -L, to -M, to -H grades, HPMCAS becomes more hydrophobic, and the minimum pH above which it dissolves increases [14]. According to Friesen et al., HPMCAS grades are sparingly soluble above pH values of 4.8 (-L), 5.2 (-M), and 5.7 (-H), dispersing to form colloidal solutions [14]. When partially ionized, hydrophobic regions of HPMCAS can interact with hydrophobic drugs, and carboxylate groups can interact with the aqueous phase at the drug–water interface to inhibit crystal nucleation and growth [14,15,16,17].

Compositions, spray solvents, and total solids loadings are shown in Table 1. Solutions were spray dried with an outlet temperature of 45–50 °C and inlet temperature of 142–150 °C on a custom laboratory scale spray dryer with a 35 kg/h drying gas capacity and a 0.3 m chamber diameter. A Schlick 1.5 pressure-swirl nozzle was used for the 25% drug loading formulations, and a Schlick 2.0 pressure-swirl nozzle was used for the 50% formulations (model 121, 150 um and 200 um orifice, Schlick Americas, Bluffton, SC, USA). After material was collected in a cyclone, residual solvent was removed by secondary drying in a vacuum dryer (Model TVO-2, Cascade TEK, Cornelius, OR, USA) for >16 h at 40 °C with a nitrogen sweep gas (−60 cm Hg, 3 standard liters per minute). Solvent removal was below the International Council for Harmonization (ICH) thresholds for methanol (<3000 ppm) as confirmed using a gas chromatograph with a headspace sampler (GC).

Powder X-ray diffraction (PXRD) was used to confirm that ASDs were amorphous, and modulated differential scanning calorimetry (mDSC) was used to ensure a single T_g_ as an indication of drug–polymer homogeneity. Scanning electron microscopy (SEM) was performed to visualize morphology and to detect the presence of any surface crystals. Method details can be found in Section A.2.

ASD immediate release (IR) tablets were made using the 50/50 (*w/w*) acalabrutinib/HPMCAS-H ASD. This ASD was chosen as the lead after dissolution testing, since it maximized both in vitro performance (i.e., AUC) and drug loading. The 50 wt % drug loading ASD resulted in ASD tablets that were 60% smaller than Calquence capsules (by volume) at an equivalent 100 mg unit dosage strength. ASD tablets had a 400 mg total tablet mass and a drug loading of 25 wt %. Refer to Mudie et al. [8] for tablet formulation, manufacturing methods, and characterization.

### 2.4. In Vitro Dissolution Testing of Intermediates

ASDs were evaluated for dissolution performance in a gastric to intestinal transfer dissolution test using a Pion µDiss^TM^ Profiler with Rainbow™ fiber optic UV probe detection. Tests using all six ASDs were first conducted using simulated gastric media with elevated pH representative of dogs (or humans) taking an ARA. While gastric pH can vary from ≈4 to 7 with ARAs, gastric pH values of 5 and 6 were chosen, since the solubilities of different HPMCAS polymer grades (-L, -M, and -H) are most sensitive in this range [18,19,20,21,22].

Relative ASD performance in the elevated gastric pH tests was used to select lead ASDs (25 and 50 wt % drug loading HPMCAS-H ASDs and 50 wt % drug loading HPMCAS-M ASD). Then, these ASDs were tested at a gastric pH representative of dogs pre-treated with pentagastrin (i.e., pH 2). Simulated intestinal medium was representative of fasted dogs, with a pH of 6.5 and 0.5 wt % (6.7 mM) FaSSIF powder (see Section A.3 for detailed medium compositions) [23,24,25].

Tests were conducted at dose concentrations of 2 (gastric) and 1 (intestinal) mg/mL, which approximated dose/volume in the stomach of fasted beagle dogs taking a 100 mg dose of acalabrutinib and resulted in ‘non-sink’ conditions (dose/volume/solubility > 1) with respect to both the apparent amorphous and crystalline solubilites in intestinal medium [26]. Refer to Section A.3 for dose number calculations.

For the tests conducted using pH 5 and pH 6 simulated gastric media, crystalline acalabrutinib and ASDs were prepared as a suspension in 0.5% Methocel A4M at a concentration of 25 mg/mL acalabrutinib. To begin the experiment, 0.8 mL of suspension was added to 9.2 mL of gastric fluid to achieve a dose concentration of 2 mg/mL acalabrutinib. For the tests conducted using pH 2 simulated gastric medium, neat crystalline or ASD powder was added directly to the dissolution vessel to begin the experiment. Samples were stirred at 100 rpm and held at 37 ± 2 °C by circulating water through a heating block mounted to the Pion µDiss™ profiler. After 30 min, gastric medium was diluted 1:1 with a concentrated intestinal medium to a final volume of 20 mL and a concentration of 1 mg/mL acalabrutinib.

Data were collected for 30 min in gastric medium and for approximately 160 min in intestinal medium with Pion Rainbow™ UV probes. See Section A.3 for UV analysis parameters. The apparent concentrations measured consisted of (1) drug dissolved in aqueous medium and (2) drug partitioned into bile salt micelles (present in intestinal medium) as a micelle-bound drug. Each sample was measured in duplicate.

AUC in intestinal medium for each test was calculated in Microsoft Excel (Microsoft Corporation, Seattle, WA, USA) using the rectangular rule using 0.25 min time increments between times 30 and 176 min of the dissolution test (i.e., for a 146 min duration in intestinal medium). AUC enhancement was determined for each ASD sample by dividing AUC in intestinal medium for the ASD by AUC in intestinal medium for crystalline acalabrutinib tested using the same dissolution conditions.

### 2.5. In Vitro Dissolution Testing of Dosage Forms

ASD tablets and commercially available Calquence capsules were evaluated for dissolution performance in a gastric to intestinal transfer dissolution test using a Vankel VK7000 (now Agilent, Palo Alto, CA, USA) United States Pharmacopeia (USP) 2 dissolution apparatus equipped with 500 mL vessels. Tests were conducted using simulated gastric media (HCl) representative of dogs taking ARAs (pH 6) or dogs pretreated with pentagastrin (pH 2). Simulated intestinal medium was representative of fasted dogs, with a pH of 6.5 and 0.5 wt % (6.7 mM) FaSSIF powder (see Section A.3 for detailed medium compositions).

Tests were conducted at dose concentrations of 0.4 (gastric) and 0.2 (intestinal) mg/mL to allow enough of the dose to dissolve in pH 2 and pH 6 gastric media, and in intestinal medium for both the ASD tablet and Calquence capsule to facilitate the determination of z-factors (i.e., dissolution rates) for in silico predictions [27]. This dose concentration allowed for ≈20–100% dose dissolved for the Calquence capsule and ≈40–100% dose dissolved for the ASD tablet across the three media. Refer to Section A.3 for dose number calculations. In vito testing at a higher, non-sink dose concentration was conducted in a controlled transfer dissolution (CTD) test as described by Mudie et al. [8].

To begin the test, 250 mL of gastric medium was added to the dissolution vessel, which was followed by a single ASD tablet, or a Calquence capsule (100 mg dose) contained in a capsule sinker. After the dosage form was added, the paddles were started. Samples were stirred at 75 rpm and held at 37 ± 2 °C by circulating water through a heater attached to the USP 2 dissolution apparatus. After 30 min, gastric medium was diluted 1:1 with a concentrated intestinal medium to a final volume of 500 mL and a concentration of 0.2 mg/mL acalabrutinib.

Dissolution performance was monitored for 30 min in gastric medium and for 150 min in intestinal medium with Pion Rainbow™ UV probes. See Section A.3 for UV analysis parameters. The apparent concentrations measured consisted of (1) drug dissolved in aqueous medium and (2) drug partitioned into bile salt micelles (present in intestinal medium) as a micelle-bound drug. All samples were analyzed in duplicate.

AUC in intestinal medium for each test was calculated in Microsoft Excel using the rectangular rule using 0.25 min time increments between times 30 and 150 min of the dissolution test. AUC enhancement was determined for the ASD tablet by dividing AUC in intestinal medium for the ASD tablet by AUC in intestinal medium for the Calquence capsule tested using the same dissolution conditions.

### 2.6. Physicochemical Property Measurements of Lead ASD Intermediate

#### 2.6.1. Amorphous Acalabrutinib Slurry pH

Slurry pH was measured as a means to estimate solid surface pH of the 50/50 (*w/w*) acalabrutinib/HPMCAS-H ASD in different HCl molarities (pH 1.6, 2.0, 3.0, 4.0, 5.0, and 6.0) containing 34 mM of NaCl [28,29]. Briefly, the 50/50 (*w/w*) acalabrutinib/HPMCAS-H ASD was added to each HCl medium and allowed to dissolve until a plateau in the pH was achieved with excess solids present. During the measurements, slurries were mixed and controlled to a temperature of 37 ± 1 °C. Measurements were performed with one sample each. Refer to Section A.4 for full method details.

Moderate intrinsic solubility of acalabrutinib coupled with its basic pK_a_s of 3.5 and 5.8 result in a solid surface pH higher than the bulk pH when solubility at the solid particle surface exceeds the buffering capacity at the solid particle surface [29,30]. Increased solid surface pH can decrease acalabrutinib solubility due to decreased acalabrutinib ionization at the solid particle surface, resulting in a decreased dissolution rate [30]. Therefore, knowledge of solid surface pH as a function of bulk pH and medium composition is critical for accurate dissolution rate predictions.

A similar measurement method was used by Pepin et al. using neat crystalline acalabrutinib [29]. For the ASD, acalabrutinib is present in the amorphous form with a higher intrinsic solubility than crystalline acalabrutinib. In addition, the ASD contains HPMCAS-H, which contains acidic groups ionizable in the GI pH range [14]. Therefore, measurements were repeated for ASDs, since amorphous acalabrutinib and HPMCAS-H in the formulation could result in a different solid surface pH than for neat crystalline acalabrutinib.

#### 2.6.2. Amorphous Acalabrutinib pH-Solubility

Solubilities of amorphous acalabrutinib dosed as the 50/50 (*w/w*) acalabrutinib/HPMCAS-H ASD were determined in different HCl molarities (pH 4.0, 4.5, 5.0, and 6.0) containing 34 mM NaCl. This approach was used rather than measuring amorphous solubility in the absence of polymer, since polymer has been shown to impact the apparent amorphous solubility [13,31]. After dosing the 50/50 (*w/w*) acalabrutinib/HPMCAS-H ASD to each medium, HCl was automatically titrated to maintain the target starting pH during dissolution using a Metrohm Titrado apparatus (Metohm, Tampa, FL, USA). During the measurements, slurries were mixed and controlled to a temperature of 37 ± 1 °C. Once a plateau in pH was achieved, indicating the end of dissolution and saturation of acalabrutinib, two aliquots of each slurry were spun down using microcentrifuge. Drug concentration in the supernatant was analyzed by high-performance liquid chromotography (HPLC). See Section A.4 for HPLC method details.

### 2.7. In Silico Predictions of In Vivo Performance

In silico predictions for acalabrutinib blood plasma versus time concentration (C_p_) profiles were performed using GastroPlus™ v9.8 simulation software (Simulations Plus Inc. Lancaster, CA, USA) with an “IR:tablet” (ASD tablet) or “IR:capsule” (Calquence capsule) dosage form setting (see detailed parameters in Table 2). All noncompartmental pharmacokinetic (PK) analyses were performed in Microsoft Excel. Values for AUC extrapolated to infinity (AUC_0-inf_), maximum drug plasma concentration (C_max_), and time to maximum plasma concentration (T_max_) were calculated for both simulated and observed plasma profiles. In silico prediction accuracy was evaluated by calculating the absolute average fold error (*AAFE*) of the predicted values for AUC_0-inf_, C_max_, T_max_, and C_p_ versus time. *AAFE* can be calculated using Equation (1), where *n* is the number of individuals and subscript *i* is the parameter of interest for determining *AAFE* (e.g., AUC) [32,33].

(1)
$$AAFE=10^{\frac{1}{n}\times \sum \left|Log\left(\frac{predicted_{i}}{observed_{i}}\right)\right|}$$


For example, an *AAFE* value of 1.3 suggests that the spread of the observed value around the predicted value is 1.3-fold, with a value of 1 having zero spread and exact agreement. An *AAFE* of <2 from a “bottom–up” in silico prediction was considered acceptable for this study.

#### 2.7.1. Physicochemical Properties and In Vitro Bioperformance Inputs

Physicochemical properties for acalabrutinib including molecular weight, pK_a_, log D, and effective permeability (P_eff_) were either referenced from prior literature studies, extracted from regulatory filing documents, or estimated from ADMET Predictor v9.5 (Simulations Plus Inc. Lancaster, CA, USA) (see Table 2) [4,6]. Solubility versus pH was referenced from Pepin et al. for the Calquence capsule and measured for the ASD tablet [6,29].

Dissolution rate in the in silico prediction was calculated using the z-factor model from the in vitro dissolution data in gastric media (see Section 3.4) for both the Calquence capsule and ASD tablet at low (starting pH 2) and high gastric pH (starting pH 6) [27]. In addition, the z-factor was also calculated from simulated intestinal medium in the presence of bile salt micelles at a pH of 6.5, using the experimental data from the pH 6 to 6.5 gastric to intestinal transfer test (see Section 3.4). Subsequently, a z-factor versus pH profile was constructed for each formulation and physiological condition (i.e., pentagastrin or famotidine pretreatment) and used for each simulation to account for changes in dissolution rate versus pH down the GI tract.

Pepin et al. reported that precipitation does not have a significant impact on acalabrutinib absorption via dissolution testing at the clinically prescribed 100 mg dose [6]. In support of this assumption, the measured in vitro dissolution data for the Calquence capsule and for the ASD tablet demonstrated no precipitation to a lower dissolved drug concentration during the duration of the experiment in the USP 2 gastric to intestinal transfer test (see Section 3.4) or in the CTD apparatus at a physiological dose concentration (see Mudie et al. [8]). The mean precipitation time in the in silico prediction was set to 100,000 seconds to reflect negligible precipitation.

#### 2.7.2. Physiology

In silico predictions were performed in GastroPlus v9.8 using default physiology settings for mean transit time, fluid volume, bile salt concentration, and small/large intestinal pH for a fasted beagle dog. Stomach pH was first adjusted to account for pentagastrin (bulk pH 2) or famotidine (bulk pH 6) pretreatment [21,22]. Subsequently, the surface pH of the dissolving solid was taken into account using the method of Pepin et al., where the bulk pH in the gastric compartment was adjusted to the estimated surface pH (i.e., slurry pH values) [6]. Doing so more accurately captures the solubility and dissolution driving force of acalabrutinib at the solid surface, resulting in a more accurate dissolution rate prediction than would be obtained using the bulk stomach pH (see Section A.5) [30,32].

In the intestinal compartments, bulk pH was not modified, since neither amorphous nor crystalline acalabrutinib should have a solid-surface pH that differs significantly from the buffered media at pH > 6 according to the recommendations by Mudie et al. [30]. Bulk and surface pH values used in the in silico predictions and rationale for maintaining bulk pH in the intestinal compartments are included in Section A.5.

The theoretical GI tract “compartment” surface-area-to-volume (SA/V) was selected as the absorption scaling factor (ASF) model in GastroPlus to scale SA/V in the different segments of the GI tract.

#### 2.7.3. PK

PK input parameters for in silico predictions are reported in Table 2. Clearance, volume of distribution, and elimination half-life for acalabrutinib in beagle dogs were calculated from the blood plasma concentration versus time profile for an acalabrutinib acidified oral solution published by Podoll et al. at a 30 mg/kg dose (see Section A.6 for calculation details) [34]. Oral rather than intravenous (IV) data were used for these calculations, since IV data in dogs were not available.

Lastly, first-pass extraction (FPE) was set to 25% based on a study by Pepin et al. where the average bioavailability (%F) was 75% when an acalabrutinib oral solution was dosed at 100 mg to beagle dogs (*n* = 12). This calculation (FPE = 100% − %F) assumes complete absorption of the oral solution, and therefore, %F less than 100 is due to metabolism in the gut and liver [6].

The assumption that acalabrutinib pharmacokinetics can be described via one central compartment was demonstrated by sufficient agreement between simulations using a one compartment model with observed data (see Section A.6).

**Table 2 pharmaceutics-13-01257-t002:** Summary of in silico prediction input parameters used for simulating blood plasma concentration versus time profiles of acalabrutinib in beagle dogs.

Drug Properties					
Parameter	Value				Source
Molecular weight (g/mol)	465.52				ADMET Predictor
pK_a_	3.5, 5.8 (basic)				Pepin et al. ^a^, FDA Biopharm review
log D (pH 7.4)	1.14				ADMET Predictor
Effective permeability (× 10^−4^ cm/s)	5.4				Pepin et al. for humans ^a^ Assumed dog permeability is similar to human
Solubility vs. pH (mg/mL)	ASD Tablet		Calquence Capsule		ASD Tablet (measured) Calquence capsule (Pepin et al. ^a^)
	pH	Solubility	pH	Solubility	
	4	6.47	4	3.9	
	4.5	2.62	5	0.34	
	5	0.81	6	0.08	
	6	0.43	7	0.05	
Biorelevant solubility—6.7 mM SIF ^b^, pH 6.5 (mg/mL)	0.71 (ASD Tablet) 0.11 (Calquence capsule)				Measured
Dissolution model (z-factor, mL/mg/s)	ASD Tablet		Calquence capsule		Calculated from in vitro dissolution data for Calquence capsule and ASD tablet (Section 3.4)
	pH	z-factor	pH	z-factor	
	4.6	0.007	4	0.001	
	6.3	0.005	6	0.03	
	6.5 (SIF ^b^)	0.011	6.5 (SIF ^b^)	0.02	
Mean precipitation time (s)	100,000				Pepin et al. ^a^
**PK Parameters in Beagle Dog (Single Compartment)**					
**Parameter**	**Value**				**Source**
Clearance (L/h/kg)	0.75				Calculated from Podoll et al. ^c^
Volume of distribution (L/kg)	1.27				
Elimination half-life (h)	1.18				
First-pass extraction (%)	25				Pepin et al. ^a^

^a^ Reference [6]. ^b^ SIF—simulated intestinal fluid comprised of 6.7 mM FaSSIF powder (Biorelevant.com, London, UK) dissolved in 67 mM phosphate-buffered saline (pH 6.5). ^c^ Reference [34].

## 3. Results

### 3.1. Dispersion Polymer Screening

Of the seven polymers screened, HPMCAS-H and HPMCAS-M showed the longest sustainment of supersaturation, with no evidence of precipitation over the 16 h test duration (see Figure 1). Eudragit L100 showed the second longest sustainment, followed by HPMC, HPMCAS-L, and PVP VA. PVP resulted in the shortest level of supersaturation sustainment at 4.8 h. Even without polymer present, acalabrutinib remained supersaturated for a relatively long duration of 3 h.

### 3.2. ASD Manufacturing and Characterization

Each spray-dried ASD displayed a single T_g_ upon mDSC measurement and showed no evidence of crystallinity (i.e., absence of sharp diffraction peaks) upon PXRD analysis. See Section A.7 for PXRD diffractograms, mDSC thermograms, T_g_ values, and SEM images.

### 3.3. In Vitro Dissolution Testing of Intermediates

ASD intermediates and crystalline acalabrutinib were evaluated in gastric to intestinal transfer dissolution tests using a Pion µDiss Profiler. All six ASDs were evaluated in tests with pH 5 and pH 6 gastric medium representative of fasted dogs taking an ARA. All six ASDs achieved higher acalabrutinib concentrations in gastric and intestinal media and higher intestinal AUCs than crystalline acalabrutinib (See Figure 2, Figure 3 and Figure 4). Whereas ASDs were supersaturated in gastric and intestinal media, low solubility of acalabrutinib at elevated gastric pH resulted in limited release in gastric medium, preventing supersaturation upon transfer to intestinal medium.

HPMCAS-H ASDs achieved the longest supersaturation durations in intestinal medium and the highest AUC enhancements (see Figure 4 and Section A.8 for tabulated values). AUC enhancements for HPMCAS-H ASDs were independent of gastric pH and drug loading within the error of the measurements. With the exception of the 25 wt % drug loading HPMCAS-M ASD tested at pH 5, which showed the second best performance, differences between AUC enhancements for the -M and -L grade ASDs were within the error of the measurements.

Since the 25 and 50 wt % drug loading HPMCAS-H ASDs achieved the highest AUC enhancements in the tests representative of fasted dogs taking an ARA, these ASDs were selected as lead candidates and carried forward into pH 2 gastric tests representative of fasted dogs pretreated with pentagastrin. The 50 wt % drug loading HPMCAS-M ASD was included as a back-up ASD. All three ASDs and crystalline acalabrutinib reached much higher extents of gastric dissolution in simulated gastric medium at pH 2 than at elevated gastric pH, and they were supersaturated upon transfer to intestinal medium (see Figure 5). It should be noted that for the gastric portion of the test, measured concentrations in gastric media may be somewhat lower than actual values due to UV detector saturation. At pH 2, the ionization of weakly basic acalabrutinib results in high crystalline solubility, driving the rapid dissolution of both crystalline acalabrutinib and ASDs. Similar performance of ASDs and crystalline acalabrutinib in intestinal medium resulted in AUC enhancements near unity (see Figure 4). Overall, in vitro testing of ASD intermediates showed that HPMCAS-H ASDs maximized performance (i.e., AUC enhancement) at elevated pH and showed equivalent performance to Calquence at low gastric pH.

### 3.4. In Vitro Dissolution Testing of Dosage Forms

ASD tablets using the 50/50 acalabrutinib/HPMCAS-H ASD and commercially available Calquence capsules were evaluated for dissolution performance in gastric to intestinal transfer dissolution tests in a USP 2 dissolution apparatus. Tests were conducted using gastric media representative of dogs taking ARAs (pH 6) or dogs pretreated with pentagastrin (pH 2) at a dose concentration lower than that used for the ASD intermediates to facilitate an adequate determination of dissolution rates for in silico predictions. The 50/50 acalabrutinib/HPMCAS-H ASD was chosen as the lead ASD, since in vitro dissolution testing of ASD intermediates demonstrated that HPMCAS-H ASDs achieved the highest AUC enhancements and because the higher drug loading minimizes tablet size.

In pH 2 gastric medium, the ASD tablet and Calquence capsule achieve similar and rapid release of acalabrutinib to a value approaching the dose concentration of 0.4 mg/mL (see Figure 6). Upon 1:1 dilution with concentrated intestinal medium, acalabrutinib concentration is halved, maintaining supersaturation at the 0.2 mg/mL intestinal dose concentration for the duration of the test.

In pH 6 gastric medium, neither the ASD tablet nor Calquence capsule show complete release of acalabrutinib in gastric medium (see Figure 6). The ASD tablet reaches a 1.8-fold higher acalabrutinib concentration in gastric medium compared to the Calquence capsule, achieving complete release upon dilution with concentrated intestinal medium and maintaining supersaturation at the 0.2 mg/mL intestinal dose concentration for the duration of the test. In contrast, the Calquence capsule is solubility-limited in both pH 6 gastric medium and intestinal medium. Acalabrutinib concentrations plateau near the crystalline solubility in both media, reaching 53% of the dose dissolved in intestinal medium by the end of the test.

Overall, in vitro testing of dosage forms showed that the ASD tablet matched the performance of the Calquence capsule at gastric pH 2 and exceeded the performance of the Calquence capsule at gastric pH 6 with an AUC enhancement of 2.2 (2.2–2.3). In vitro testing at a higher dose concentration in an in vitro CTD test demonstrated similar relative performance between dosage forms at a given initial gastric pH (see Mudie et al. [8]).

### 3.5. Physicochemical Property Measurements of Lead ASD Intermediate

Slurry pH and amorphous solubility values of the 50/50 (*w/w*) acalabrutinib/HPMCAS-H ASD were measured at different HCl concentrations (i.e., bulk pH values). Results are shown in Table 3. Molar concentrations of acalabrutinib in solution are high enough to increase slurry (ending) pH above the starting values, and they are higher than the values reported by Pepin et al. for crystalline acalabrutinib due to the higher solubility of the ASD [29]. The measured amorphous solubilities shown in Table 3 are higher than crystalline solubilities reported by Pepin et al.; for example, they are 5.6-fold higher at pH 6.0 where acalabrutinib is ≈63% non-ionized [29].

### 3.6. In Silico Predictions Versus In Vivo Performance

The in silico PK predictions for acalabrutinib are shown in Figure 7 with tabulated results comparing simulated versus observed data in Table 4. The simulations provided good prediction accuracy of the observed data. The AAFE of the in silico predictions for AUC_0-inf_, C_max_, T_max_, and C_p_ versus time for all formulation treatments were <2-fold (ideal value = 1) with the exception of C_p_ versus time for the Calquence capsule + famotidine treatment, indicating that the in silico prediction framework is sufficient for simulating acalabrutinib blood plasma concentrations within this in vivo study.

## 4. Discussion

### 4.1. Benefits of Streamlined, In Vitro–In Silico ASD Development Approach

ASD dosage forms are useful for increasing the bioavailability of poorly soluble drugs, thereby enabling effective oral delivery and improving patient safety and compliance, such as by removing DDIs, reducing plasma variability, reducing dose, and mitigating food effects [35]. As highlighted in this study, ASDs can be effective for formulating weakly basic drugs, which make up 78% of new molecular entities approved between 2003 and 2013 that showed a clinical DDI with ARAs [3].

To be robust, ASD dosage forms must achieve good in vivo performance, stability, and manufacturability, while minimizing dosage form size. Attaining all of these attributes can be challenging, often resulting in numerous formulation iterations that increase development timelines and costs. As described in this study, knowledge of key drug, polymer, and GI properties can be combined with in vitro and in silico tools to design ASD formulations that achieve good performance in preclinical studies, thereby helping to mitigate these issues.

This streamlined ASD development approach not only achieved the in vivo study goal on the first try with a single ASD tablet formulation but also minimized the time and amount of drug needed for formulation development. For example, this approach included the use of a high-throughput, material-sparing solvent-shift UV test, which identified the most effective dispersion polymers prior to spray-drying ASDs, reducing the number of ASD formulations. Next, Pion µDiss in vitro dissolution testing was used to screen performance and select the lead ASD intermediate, before making tablets, allowing for minimal use of drug and ASD material while achieving high throughput. Once the lead ASD was identified, a tablet dosage form was made of this ASD alone, which was evaluated against the Calquence capsule in a USP 2 dissolution apparatus, as well as a physiologically relevant CTD apparatus (as described by Mudie et al.) [8]. Finally, in vitro data together with drug physicochemical and physiological properties were input into in silico software to maximize the likelihood of the ASD tablet working as intended vivo.

Throughout this approach, knowledge of the problem statement (i.e., pH-dependent absorption of a BCS 2 weak base) along with key drug, polymer, and physiological properties facilitated effective design of in vitro testing methods and media compositions. For example, gastric to intestinal transfer and controlled transfer tests were used to evaluate the precipitation of a weakly basic drug, and gastric and intestinal media were selected to test performance over a pH range relevant to dogs treated with pentagastrin and ARAs. Further, nuances of acalabrutinib, such as surface pH effects on dissolution rate were accounted for in the in silico simulations. Taken together, this streamlined approach resulted in successful removal of the Calquence ARA effect in a beagle dog model without the need for multiple in vivo studies or reformulation. In addition, ASD tablets were 60% smaller than Calquence capsules, and they achieved good stability and manufacturability, as described by Mudie et al. [8].

### 4.2. Relative In Vitro Performance of HPMCAS ASDs

This study examined the effects of HPMCAS grade and drug loading on the in vitro performance of acalabrutinib ASDs. While all ASD intermediates achieved higher AUCs than crystalline acalabrutinib in intestinal medium (see Figure 2, Figure 3 and Figure 4), HPMCAS-H ASDs achieved the highest intestinal AUC enhancements of the three grades. These highest AUC enhancements were due to the longest supersaturation durations in intestinal medium, which was potentially due to having the highest degree of hydrophobicity and therefore best crystallization inhibition of the three grades, as shown for some drugs [15,36,37].

While intestinal AUC enhancement was the dominant metric used to select the lead ASD, the extent of gastric dissolution varied with polymer grade, drug loading, and initial gastric pH. Notwithstanding these differences, all -M and -H ASDs dissolved nearly instantaneously to the apparent amorphous solubility in intestinal medium representative of dogs, with AUC enhancements for the lead, 50 wt % drug loading HPMCAS-H ASD ≥ 2 fold greater than for all other ASDs. The goal of the study was to develop an ASD tablet that could achieve equivalent AUC at low and high gastric pH and match the performance of Calquence as prescribed (i.e., at low gastric pH). Indeed, this metric was achieved in a dog model described herein and by Mudie et al. [8].

### 4.3. In Silico PK Simulation Tools in Early Development

In silico PK simulations are emerging tools for ASD formulation development, since they can simulate many dynamic physiological factors that impact ASD bioperformance [13,38,39,40]. Simulations using physiologically-based pharmacokinetic (PBPK) modeling software such as GastroPlus can help design and optimize in vivo studies with respect to dose, prandial state, or pH modification (i.e., selecting ARAs for pH effect study) to maximize the likelihood of achieving desired ASD in vivo performance.

Since ASD formulations are typically first evaluated in a preclinical setting, early in silico predictions in the absence of a validated biomodel are important for understanding formulation or physiological parameter sensitivity and rank ordering formulations. In fact, hypothesis testing via in silico predictions connects all aspects of ASD formulation selection from excipient screening to in vitro dissolution of the target dosage form—providing appropriate affirmation at each stage of development.

For acalabrutinib, in silico prediction was implemented early in formulation evaluation to aid in vivo study design and execution and to understand parameter sensitivity (solubility and dissolution rate as a function of pH) for lead formulations. Predictions provided confidence that the 100-mg dose of Calquence capsules in dogs would show reduced exposure (i.e., a pH effect) with the famotidine (high stomach pH) pretreatment compared to the pentagastrin (low stomach pH) pretreatment. Importantly, the predictions also forecasted that the ASD tablets could overcome the pH effect by showing equivalent AUC across both pentagastrin and famotidine pretreatments, highlighting their superior performance to the Calquence capsule. Therefore, the in silico predictions performed herein were an integral and necessary step to the successful in vivo result.

## 5. Conclusions

This study describes a streamlined in vitro and in silico development approach for designing an ASD tablet to overcome the pH-dependent absorption of the weak base drug, acalabrutinib. The methodologies described in this study can be applied directly to the development of ASDs of other weak base drugs for mitigating gastric pH effects. Furthermore, the general framework can be applied to developing ASDs of other poorly soluble drugs for bioavailability enhancement. When designed with key drug, polymer, and gastrointestinal properties in mind, in vitro and in silico tools can help identify the in vivo barriers to absorption and assess whether they can be overcome by an ASD drug product. As this approach achieved the in vivo study goal on the first try with a single ASD tablet formulation, it has the potential to reduce time and cost to introducing robust ASD drug products to the market.

## Figures and Tables

**Figure 1 pharmaceutics-13-01257-f001:**
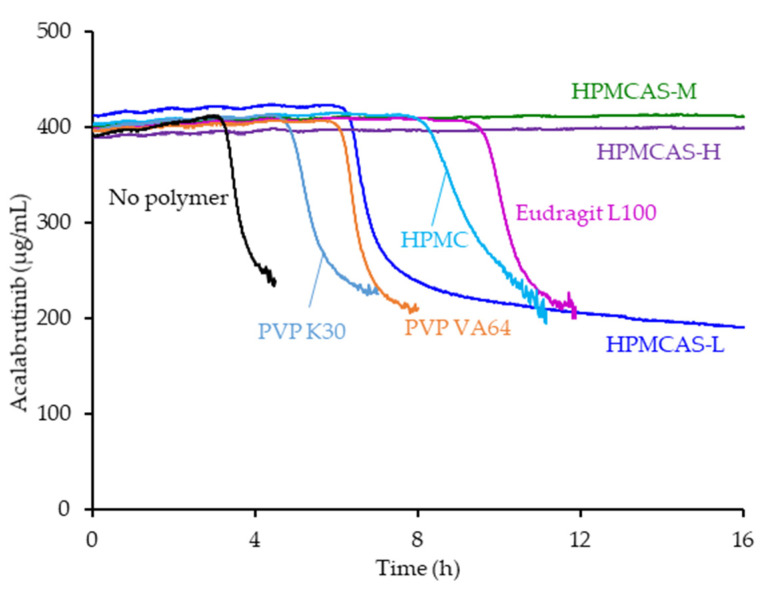
Acalabrutinib concentration versus time in pH 6.5 phosphate buffer containing 6.7 mM FaSSIF powder and 82 mM NaCl measured in the absence and presence of 200 µg/mL dissolved polymer.

**Figure 2 pharmaceutics-13-01257-f002:**
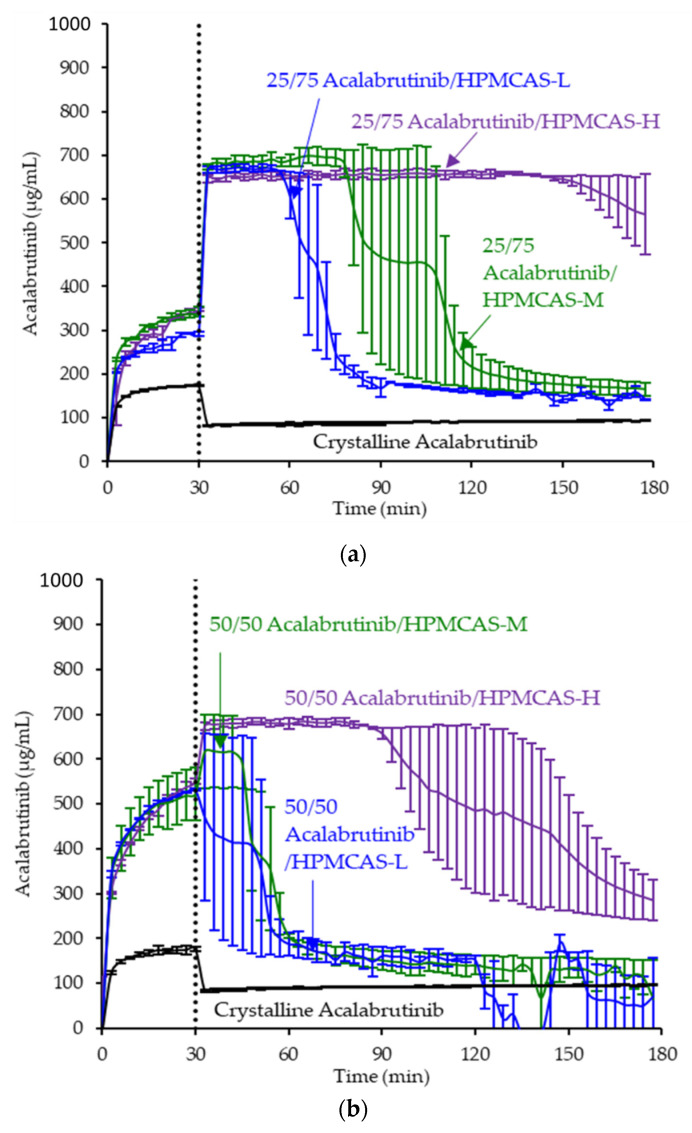
Dissolution profiles of ASD intermediates in a gastric to intestinal transfer dissolution test using a Pion µDiss Profiler under conditions simulating fasted dogs treated with an ARA (gastric pH 5) (curves represent average values whereas error bars represent range, *n* = 2). Panel (**a**) 25 wt % drug loading ASDs. Panel (**b**) 50 wt % drug loading ASDs.

**Figure 3 pharmaceutics-13-01257-f003:**
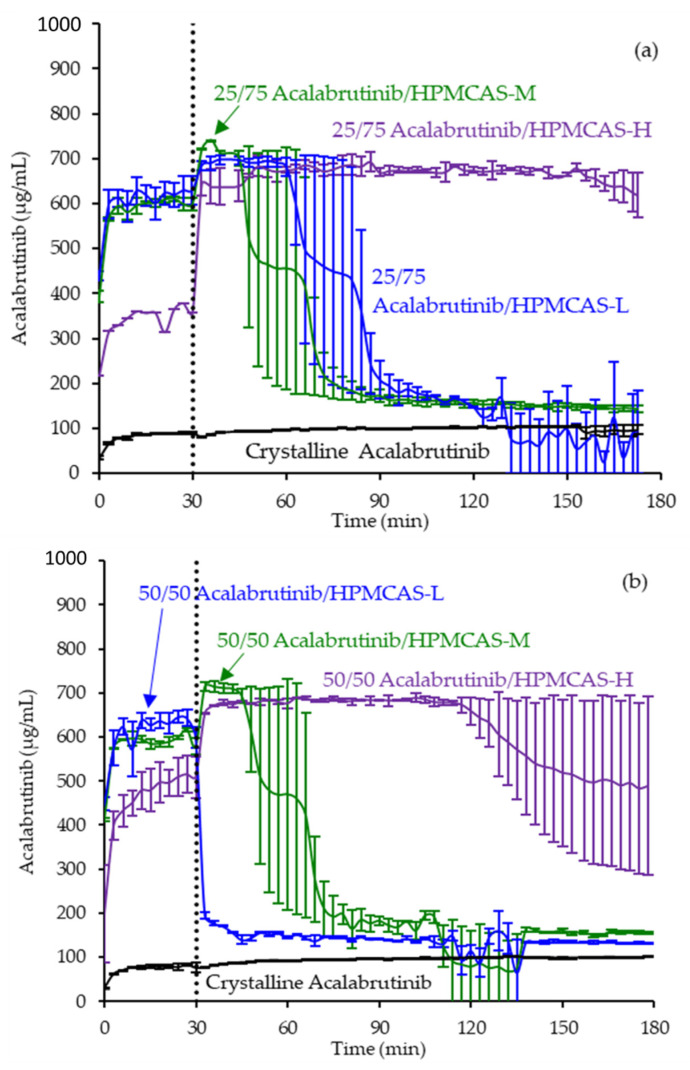
Dissolution profiles of ASD intermediates in a gastric to intestinal transfer dissolution test using a Pion µDiss Profiler under conditions simulating fasted dogs treated with an ARA (gastric pH 6) (curves represent average values whereas error bars represent range, *n* = 2). Panel (**a**) 25 wt % drug loading ASDs. Panel (**b**) 50 wt % drug loading ASDs.

**Figure 4 pharmaceutics-13-01257-f004:**
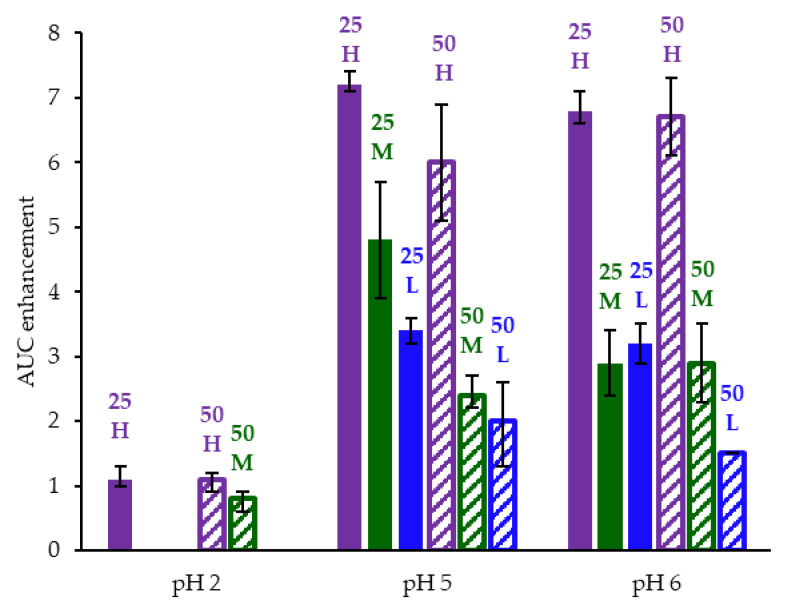
AUC enhancement for ASD intermediates tested in gastric to intestinal transfer dissolution tests using a Pion µDiss Profiler. *X*-axis labels represent starting gastric pH. AUC enhancement = AUC(ASD)/AUC(crystalline acalabrutinib), reported as average with error bars representing the range of two measurements.

**Figure 5 pharmaceutics-13-01257-f005:**
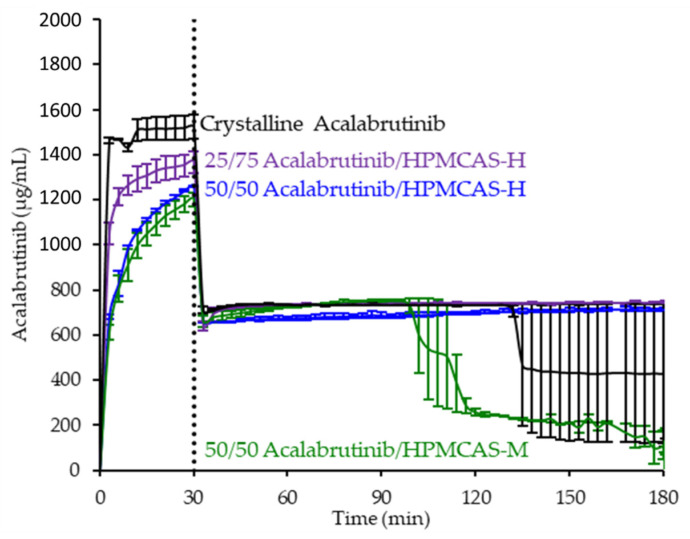
Dissolution profiles of ASD intermediates in a gastric to intestinal transfer dissolution test using a Pion µDiss Profiler under conditions simulating fasted dogs treated with pentagastrin (gastric pH 2) (curves represent average values whereas error bars represent range, *n* = 2).

**Figure 6 pharmaceutics-13-01257-f006:**
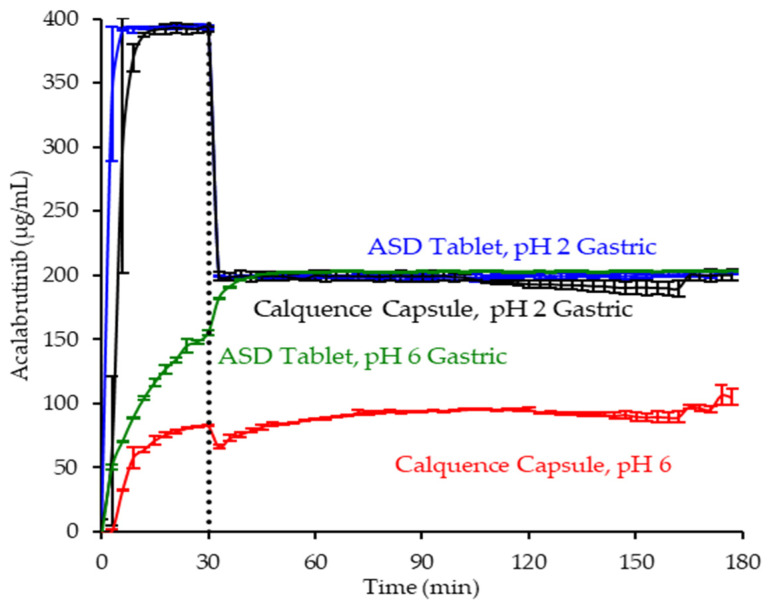
Dissolution profiles of ASD tablet and Calquence capsule in gastric to intestinal transfer dissolution tests in the USP 2 apparatus under conditions simulating fasted dogs treated with pentagastrin (gastric pH 2) or an ARA (gastric pH 6) (curves represent average values, whereas error bars represent range, *n* = 2).

**Figure 7 pharmaceutics-13-01257-f007:**
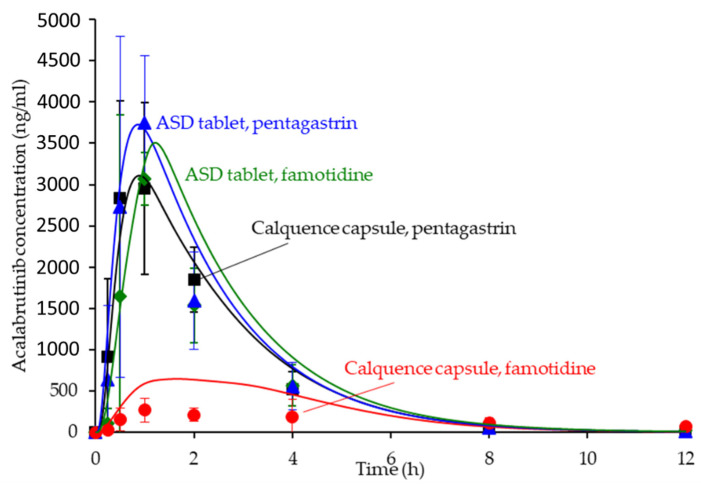
Simulated versus observed acalabrutinib blood plasma concentrations versus time at 100 mg dose in beagle dogs with pentagastrin or famotidine pretreatment. Markers are the observed average data points with error bars representing the standard deviation, and solid lines are the in silico predictions.

**Table 1 pharmaceutics-13-01257-t001:** Compositions of ASDs and spray-drying solutions.

Drug Loading (wt %)	Dispersion Polymer	Spray Solvent	Total Dissolved Drug and Polymer in Spray Solvent (wt %)
25	HPMCAS-H	Methanol	6
25	HPMCAS-M	Methanol	6
25	HPMCAS-L	Methanol	6
50	HPMCAS-H	Methanol	5.4
50	HPMCAS-M	Methanol	5.4
50	HPMCAS-L	Methanol	5.4

**Table 3 pharmaceutics-13-01257-t003:** Slurry pH and solubilities of amorphous acalabrutinib dosed as the 50/50 (*w/w*) acalabrutinib/HPMCAS-H ASD in different molarities of HCl containing 34 mM NaCl (*n* = 1).

Starting pH	Slurry (Ending) pH	Amorphous Acalabrutinib Solubility (mg/mL) ^a^
1.6	4.4	-
2.0	4.6	-
3.0	5.4	-
4.0	6.2	6.47 (6.43–6.51)
4.5	-	2.62 (2.48–2.76)
5.0	6.3	0.81 (0.80–0.82)
6.0	6.3	0.43 (0.43–0.44)

^a^ Average (range of two measurements per one sample).

**Table 4 pharmaceutics-13-01257-t004:** Noncompartmental analysis comparing simulated (sim) versus observed (obs) data for all formulation treatments in the dog study. Absolute average fold error (AAFE) was calculated for AUC_0-inf_, C_max_, T_max_, and C_p_ versus time to determine the accuracy of the in silico prediction exercise (ideal value = 1).

Formulation	AUC_0-inf_ (ng h/mL)		C_max_ (ng/mL)		T_max_ (h)		AAFE			
	Obs	Sim	Obs	Sim	Obs	Sim	AUC_0-inf_	C_max_	T_max_	C_p_ vs. Time
ASD tablet, pentagastrin	8161	9766	3332	3727	0.9	0.9	1.2	1.2	1.6	1.3
ASD tablet, famotidine	7579	9555	3443	3508	0.9	1.6	1.3	1.2	1.8	1.6
Calquence capsule, pentagastrin	8365	8607	4480	3110	0.8	0.9	1.1	1.4	1.3	1.3
Calquence capsule, famotidine	3112	3096	355	648	1.6	1.2	1.6	1.9	1.7	3.0

## Data Availability

The data in this study in the form of Microsoft Excel worksheets are available from the corresponding author upon request.

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
