# Peer review of "In Vitro-In Silico Tools for Streamlined Development of Acalabrutinib Amorphous Solid Dispersion Tablets"

_pharmaceutics, 2021, doi:10.3390/pharmaceutics13081257_

Round 1

Reviewer 1 Report

MS title: In vitro-in silico tools for streamlined development of acalabrutinib amorphous solid dispersion tablets

The work done is sufficient and attentive to the objectives. Experimental design and studies are described in detail. The achieved results are relevant and the interpretations of results are well-done and are based on specific literature. However, few points that would be improved for the publication are as under. 

A similar article has already been published,  Amorphous Solid Dispersion Tablets Overcome Acalabrutinib pH Effect in Dogs , Pharmaceutics. 2021 Apr; 13(4): 557. Authors should comment, the novelty of submitted article with the published one. A similar PK values presented in Table Table 4 are already reported in Table A5 of published article.

Authors should mention important finding in abstract. Remember that the abstract may be all readers will see if they are using an abstracting service such as PubMed, Web of Science, etc. as a way of finding relevant papers.

In the conclusion section, don’t just repeat the results. Indicate if the study design can be generalized to a broader study population and the possibilities for future research.

Graphs of DSC, XRD, FTIR etc should be included while revising the manuscript, showing compatibility of ingredients of tables. 

Authors should comments on stability issue of tablets. 

What about the validation of HPLC method? What is the r2 and other validation parameters of HPLC. HPLC Chromatograms should be added.

The Y-axis scale of figure 2 and 3 should be adjusted according to the values of drug concentration.

*Correct the italic style of ‘in vitro and ex vivo’ in all the text.

Authors need to revise figures legend. it is not stated in the manuscript that the results are expressed either in "Mean +/- Standard deviation (SD)" or "Mean +/- Standard error mean (SEM).

In the references: Update the doi of published articles with the volume and pages, check the actual journal if necessary.

Author Response

Please see the attached point-by-point response to reviewer 1.

Reviewer 2 Report

The manuscript "In vitro-in silico tools for streamlined development of acalabrutinib amorphous solid dispersion tablets" by Mudie et al. describes a process of development of amorphous solid dispersion composed of acalabrutinib.

First the authors presents preliminary studies, which led to selecting right  polymer and spray drying process parameters. Then the authors focused on the in vitro assays describing the dissolution performance of ASD compared with commercially-available Calquence. 

As a proof of concept the authors proceed with the development of in silico PK model and they measured the physicochemical properties of obtained ASD.

The manuscript is an example of properly conducted research without major concerns. However, there are few questions which arise from the lecture of the manuscript:

1) How were the tables manufactured? And how they can be characterized? Please provide the information on the tablet press, dimension of tablets, compression force, hardness, friability, composition (or if the mixture was plain 50/50 (w/w) acalabrutinib/HPMCAS-H ASD),

2) What were the results of stability studies? The authors claim that the ASD tablets have good stability when they were stored refrigerated or desiccated, but no results were presented. Moreover, in my opinion the authors should rather focus on stablizing the ASD to perform better when stored in room temperature, where the most of the tables are stored, then moving forward with in vivo studies.

 In the light of above I recommend the manuscript to be published in the Pharmaceutics after major revision.

Author Response

Please see the attached point-by-point response to reviewer 2.
